# Who Benefits from the Fixed Copayment of Medical and Pharmaceutical Expenditure among the Korean Elderly?

**DOI:** 10.3390/ijerph17218118

**Published:** 2020-11-03

**Authors:** Eunja Park, Sookja Choi

**Affiliations:** 1Korea Institute for Health and Social Affairs, Sejong 30147, Korea; ejpark@kihasa.re.kr; 2Red Cross College of Nursing, Chung-Ang University, Seoul 06974, Korea

**Keywords:** fixed copayment, medical expenses, Korea, pharmaceutical expenditure, elderly

## Abstract

The Korean National Health Insurance system imposes a 30% coinsurance for outpatient medical care and prescription drugs; however, at the age of 65, the coinsurance model changes to a copayment model that offers lower fees for the elderly. Thus, this study aimed to investigate the influence of the copayment model for outpatient visits and prescription drugs on healthcare utilization among the Korean elderly. We compared total outpatient visits, total prescriptions, and out-of-pocket expenses between a case group with copayment reduction (65 years or older) and a control group without any reduction (64 years or younger). We obtained secondary data collected from seven waves of the Korea Health Panel Survey (2010–2016). Outpatient visits increased exclusively in the case group among those with lower income. After adjusting for covariates, the results of the difference-in-differences analysis showed that, compared to the control group, there was a significant increase in outpatient visits among individuals with lower income in the case group. Our study shows that cost sharing changes affect Korean patients with different income levels in different ways.

## 1. Introduction

With the rapid increase in healthcare expenditure, increased international attention has been drawn toward policy measures aimed at controlling the financial burden of healthcare [1,2]. In this context, cost sharing mechanisms are frequently deemed as potential solutions to the rapidly increasing cost burden faced by many universal healthcare systems [3,4,5,6], which can prevent users from over utilizing healthcare services. It limits the insured moral hazard (i.e., unnecessary use of services covered by health insurance) by making patients consider the necessity of utilizing their health insurance [3,7]. However, cost sharing has clear trade-offs; although it can reduce direct costs by mitigating the risk of moral hazard associated with the use of healthcare services, it may also reduce access to beneficial and essential healthcare that could mitigate future severe and expensive health issues [8]. Additionally, the financial protection function of a health insurance system may be diminished or weakened if the cost sharing policy lacks generosity [9]. Hence, policy decisions on cost sharing must consider the pros and cons of the intervention [10]. Previous studies have explored the variation in healthcare demand owing to changes in cost sharing policies. Some studies reported that in insured populations, an increase in, or introduction of, cost sharing reduced healthcare expenditure and service utilization [4,11,12,13], while others showed that service utilization increased as copayments decreased [14,15]. Nevertheless, in these studies, policy changes had significant effects on reducing the patients’ out-of-pocket (OOP) expenses for outpatient and emergency services.

Korea’s National Health Insurance (NHI) program is a universal and compulsory health insurance scheme that covers 97% of the Korean population [16]. Moreover, South Korea allows patients to choose freely between medical care organizations, and has a payment system based on fee-for-service [17,18,19,20]. In 2007, the copayment model, in which patients paid a fixed amount per visit, was replaced by the coinsurance model where patients pay a proportion of the cost; hence, the NHI system now imposes a 30% coinsurance for outpatient medical care and prescription drugs [20,21]. This policy change was aimed at reducing the use of the NHI system by the non-elderly for minor illnesses (e.g., a common cold), which occurred frequently in the outpatient setting owing to an increase in the coverage of OOP expenses by the NHI program [19,21].

The elderly are heavily dependent on healthcare services for the treatment of various illnesses and are economically vulnerable because they are retired [17]. Therefore, in this new system, after people turn 65 years old, the coinsurance model changes to a copayment model with lower fees (hereinafter lower copayment model) [19,21]. For example, if an outpatient visit costs less than 15,000 won (US $14), patients who are 65 years or older are required to pay only 1500 won (US $1.4); similarly, if a prescription drug costs less than 12,000 won (US $11), elderly patients pay only 1000 won (US $0.9). Notwithstanding the above, when the visit or prescription drug exceeds the expenditure limit, elderly patients must pay the same 30% coinsurance as non-elderly patients. The elderly are considered a vulnerable population. Hence, this policy change aims to protect them from financial burden due to higher cost sharing fees [3,6].

Despite the aforementioned importance of policy decision making concerning the reduction in copayments to improve healthcare utilization, few studies have examined the impact of policy changes in ambulatory or outpatient care settings [15]. A previous study [22] also acknowledges this gap in the literature. It showed that, although healthcare is important for the elderly, the influence of cost sharing on healthcare utilization has yet to be thoroughly investigated in elderly patients. Existing studies have evaluated the impact of the aforementioned Korean policy change either by analyzing cross-sectional data or using time series analysis [19,21]. Using data from 2007–2008, Bae et al. [17] compared healthcare utilization between Korean adults aged 65–69 years and those aged 60–64 years. However, whether or not this policy directly influences patients’ behavior has yet to be determined. Instead, most research to date has examined the effects of cost sharing on healthcare utilization by analyzing aggregated data from non-elderly patients [7,17,18,19,23,24]. To bridge this gap, we focused on the behavioral changes of Korean elderly patients due to reductions in cost sharing fees.

To evaluate the causal effects of this Korean policy and its subsequent changes, randomized controlled trials or quasi-experimental methodologies are preferred, as they use treatment and control groups to investigate between-group outcome differences. Hence, by utilizing a quasi-experimental design, we examined changes in healthcare utilization before and after patients were affected by this policy. Specifically, we analyzed between-group longitudinal changes in the use of healthcare services due to the health insurance coverage model changing from coinsurance to (lower) copayments for outpatient medical care and prescription drugs in elderly patients. We also evaluated the effects of the lower copayment model, according to income level.

Therefore, the aim of this study was to investigate the influence of the health insurance copayment model for outpatient visits and prescription drugs on healthcare utilization among the Korean elderly. We compared total outpatient visits, total prescriptions, and OOP expenses between a case group that used the lower copayment model and a control group before and after the age of 65—the age at which the copayment reduction begins to take effect.

## 2. Materials and Methods

We obtained secondary data collected from seven waves of the Korea Health Panel (KHP) survey (2010–2016). Since 2008, the KHP survey has been conducted annually to examine healthcare utilization and expenditure among community-dwelling populations nationwide [25].

The new lower copayment model for outpatient medical care and prescription drugs is only accessible to patients who are 65 years or older. Accordingly, the inclusion criterion for the case group was age—65 years in the 2011, 2013, and 2015 KHP surveys. The patients in the case group experienced a change in copayments after their 65th birthday in 2011, 2013, and 2015. The inclusion criterion for the control group was age 64 years in each of the three surveys without a change in copayments after their birthday in 2011, 2013, and 2015.

This cost sharing policy for the elderly was exclusive to NHI beneficiaries. The exclusion criterion for both groups was being a beneficiary of other types of health insurance in Korea. In total, the study population comprised 558 participants in the case group and 587 participants in the control group. This study was approved by the Institutional Review Board of Korea Institute for Health and Social Affairs (IRB No: KIHASA 2017-17).

In this study, the case group was coded as 1 and the control group was coded as 0. To determine whether healthcare utilization data came from participants in the pre- or post-65/64 age groups, the same two numbers were used for the time variables (i.e., 1 = post-65/64 and 0 = pre-65/64 years). The influence of the copayment model on healthcare utilization among the elderly was measured by the interaction of the dummy group and time variables.

In the KHP survey, we examined healthcare utilization and drug prescription filling for each visit by analyzing receipts and a diary written by each panel household [25,26]. Outpatient visits referred to the total number of visits for outpatient medical services in clinics; total prescriptions referred to the total number of prescriptions that patients were prescribed during their outpatient visits one year before and after their 65th/64th birthday. For example, if one’s 65th or 64th birthday was on 1 June 2011, the pre-65/64 years outpatient visits variable was measured as the total number of outpatient visits between 1 June 2010 and 31 May 2011 and the post-65/64 years outpatient visits variable as that between 1 July 2011 and 30 June 2012. Outpatient OOP expenditure was calculated by totaling the patients’ outpatient visit or prescription drug expenses during one year.

Several studies have shown that patients with low income are vulnerable to cost sharing increases [27,28]. Therefore, we considered household income status as one of the main explanatory variables in our study. Regarding income, the KHP survey examined not only participants’ wages, salaries, and the income of self-employed family members, but also their social security income, interest, and household income. Thence, based on previous literature [29], we calculated the equivalent household income by dividing it by the square root of the number of family members. Household income status was grouped using terciles (i.e., low, middle, and high).

The control variables included sex, residential area, educational attainment, the number of chronic diseases (diagnosed by a physician), smoking status, household income, and survey year. Sex was classified as male/female; residential area as metropolitan/non-metropolitan; educational attainment as elementary school or lower/middle school/high school or higher; and the number of chronic diseases as 0–1/1/2/≥3. Smoking status was assessed by asking the question “Do you smoke now?” Participants were classified as being a current smoker, an ex-smoker, or a non-smoker based on their responses.

We presented the participants’ characteristics using descriptive analysis, and healthcare utilization was compared before and after the cost sharing change by group. Difference-in-differences (DID) analysis was performed to assess the influence of cost sharing changes. First, the difference in outpatient visits during one year before and after the cost sharing change was calculated. Then, the differences between the case and control groups were compared while adjusting for the aforementioned control variables.

The DID regression model used in this analysis can be expressed as:y = β_0_ + β_1_ • case group_j_ + β_2_ • post-copayment change_ij_ + β_3_ • case group_j_ × post-copayment change_ij_ + β_4_ • control variable_ij_ + ε_ij_. (1)

The DID analysis was performed for OOP medical care expenditure, total prescriptions, and OOP prescription drug expenditure using the same regression models. Next, we repeated the analysis by income group to test whether the influence would be similar among patients with different income levels. Statistical analyses were performed using SAS 9.3 (SAS Institute, Cary, NC, USA). Statistical significance was set at a two-tailed *p*-value of <0.05.

## 3. Results

There were no significant differences in sociodemographic characteristics between the case and control groups, except for household income. In the control group, the proportion of participants in the high income tercile was 37.7%; 7% higher than that in the case group (Table 1).

Although outpatient visits increased in the control group, it was higher in the case group, especially among those with low income. In the latter, the mean increased from 18.6 days before one’s 65th birthday to 20.9 days afterwards; in the former, the difference before and after one’s 64th birthday was very small (0.4 days). Those with low income increased their outpatient visits by 3.9 days (from 18.5 to 22.3 days) after their 65th birthday (i.e., after reduced OOP medical care expenditure).

Similarly, the mean number of prescriptions increased significantly in the case group among those with low income, from 11.9 to 13.3. However, OOP prescription drug expenditure did not demonstrate an improvement in any group. The mean OOP prescription drug expenditure increased by 3100 won (US $3) (from 119,000 to 120,000 won) after the age of 65 in the case group and by 4400 won (US $4) (from 113,000 to 116,200 won) in the control group (Table 2).

Table 3 and Table 4 show the results of the DID analysis for outpatient visits by household income. After adjusting for covariates, the group × time interaction coefficient for outpatient clinic visits was 1.83 among all participants (marginal significance; *p* = 0.062) and 4.58 (*p* < 0.01) among those with low income. Male sex (beta = −8.67; *p* = 0.005), higher educational attainment (middle school: beta = −4.91, *p* = 0.029; high school or higher: beta = −4.48, *p* = 0.069), and non-smoking status (beta = −7.82; *p* = 0.024) were negatively associated with outpatient visits among the Korean elderly with low income. Conversely, the presence of multiple chronic diseases (*n* = 2: beta = 6.53, *p* = 0.022; *n* = ≥3: beta = 15.45, *p* = <0.0001) was positively associated with outpatient visits among the Korean elderly with low income.

The group × time interaction coefficients for total prescriptions, OOP outpatient visit expenditure, and OOP prescription drug expenditure were not statistically significant in either the total study population or the subgroups stratified by income status.

## 4. Discussion

Cost sharing prevents unnecessary healthcare utilization and ensures a sustainable health insurance system [30]. It has also been used as a viable policy-related measure to change patients’ behaviors regarding healthcare insurance services. In this study, we examined the impact of the Korean fixed copayment policy for the elderly (aged 65 years and older) on their healthcare service utilization. Overall, we showed that a reduction in copayments was associated with an increase in outpatient clinic visit frequency among elderly patients with a low income.

In the case group, outpatient visits and total prescriptions increased, which may be attributed to the reduced copayment rates. There was no increase in the patients’ OOP expenditure. Similarly, Na et al. [20] reported that the elderly who benefited from a fixed copayment policy visited their doctors more frequently and had lower OOP medical expenses. Moreover, our results are consistent with those of two Japanese studies [31,32] that found that decreased copayments increased the number of doctors’ visits. The methodology of Shigeoka et al. [32] was comparable to ours, in that only short-term effects could be observed.

Furthermore, we found that only poor elderly patients were influenced by the reduced copayment policy. Potential explanations for these aforementioned behaviors among the poor Korean elderly due to policy changes are outlined hereinafter. First, we believe that the heterogeneous price responsiveness of the poor Korean elderly may depend on the proportion of their total monthly income that healthcare expenditure occupies. Regarding the price elasticity of medical demand, a study [32] suggested that the price elasticity of the elderly may be higher than that of the non-elderly, if they are poorer or more credit-constrained, or lower if their health problems are more severe. However, another study suggested that, a priori, it is unclear whether the elderly have a higher or lower price elasticity with respect to healthcare services than the non-elderly [33]. Hence, although the elderly are among the most common and intensive consumers of healthcare services, credible evidence on the price elasticity of this group remains scarce. In light of this discussion, our findings suggest that only the poor Korean elderly are more sensitive to the price elasticity of outpatient healthcare services.

A second, more plausible, explanation is that cost sharing may be a potential barrier to obtaining healthcare, especially if the OOP expenditure represents a significant proportion of a patient’s income. For example, the same amount of OOP expenditure may represent a large proportion of a poor patient’s income, while representing only a small proportion of the income of middle- or high-earning patients; this increases the financial barrier to healthcare that disproportionately affects low income patients [34]. Generally, OOP payments in the Korean NHI system are regressive; values depend on healthcare utilization, not on the patients’ income. Hence, OOP payments tend to pose a greater financial burden on the poor. Poorer households have also experienced a large increase in the proportion of healthcare expenditure relative to the total household income [35]. Moreover, the poorer elderly have a far greater number of chronic ailments; thus, their outpatient visits are more frequent and they require prescriptions more often than younger or wealthier elderlies, potentially making poorer elderlies view medical usage as a necessary service [36].

In the Korean healthcare system, clinical physicians do not act as gatekeepers but as service providers, implying that high-earning Koreans are able to visit physicians more frequently and to incur more costly healthcare services. This supposition corroborates our findings concerning high-earning elderlies in the control group: their OOP outpatient visit expenditure increased as the number of visits increased. This can be attributed to the moral hazard concept, which is characteristic of higher income groups; they often subscribe to supplementary private health insurance companies to protect against the increased medical costs, especially for very expensive services (i.e., new drugs and the latest medical technology) [37,38].

Some studies have shown that reduced copayment fees lead to increased healthcare utilization [13,39,40,41]. The impact of cost sharing is significant in outpatient and emergency services [23,39,40,42]. However, its effects are not always positive. The decrease in medical service pricing through the reduction of cost sharing fees may evoke moral hazard behaviors from patients and medical service providers alike. For example, physicians may encourage people to utilize healthcare services, even for minor symptoms. Yet, some studies suggest that increased healthcare utilization can have a positive effect on healthcare access, especially among low income populations [23,43,44]. In this study, policy changes increased only healthcare utilization, not OOP expenditure. The original cost sharing fees may have been barriers to healthcare services for some elderly patients owing to the perceived financial costs that they may incur. Furthermore, increases in healthcare utilization after policy changes may have been driven by both physicians and patients. Given that the increased outpatient visits and prescriptions occurred exclusively among the poor Korean elderly, it may be that increased healthcare utilization has indeed helped to fulfill the unmet healthcare needs of the poor Korean elderly (i.e., those who desire and need it most). We believe that a lower copayment cost sharing model may improve access to healthcare and prescription drugs that would not normally be available to the poor Korean elderly (prior to the policy change) because of higher OOP expenses.

We further surmise that the change in policy may have increased the accessibility of healthcare services among elderly patients with low income. Based on the KHP analyses, Jung and Huh [35] remarked that approximately 20% of the respondents reported having unmet healthcare needs. Furthermore, they showed that the incidence of unmet needs was higher among those aged older than 80 years, the lowest and highest income quintiles, patients with disabilities, and those with fewer academic qualifications. Based on three years of the KHP, Ko [45] showed that more than 15% of those aged 18 years and older experienced unmet healthcare needs (In 2009, 21.06%; 2011, 14.5%; and 2012, 15.75%). This study also showed that the association between participants’ economic characteristics and their unmet healthcare needs was stronger than that between any other variables and unmet healthcare needs [45].

Physician-induced demand should not be ruled out as another explanation of the aforementioned rise in healthcare utilization among the poor Korean elderly. According to Arrow [46], owing to their information advantage, physicians can influence the demand for their services to a greater extent than other professionals. Providers are also less vulnerable to financial risk burden than patients because they can pool treatment cases. In contrast, Johnson et al. [47] argued that outpatient healthcare utilization is influenced less by the physicians’ recommendations and disease severity and more by inpatient healthcare utilization. Specifically, the physicians’ role in the decision making process regarding healthcare service utilization was less important than that of the patient. In support of this, a study showed that cost sharing for outpatient healthcare utilization was not affected by the physician’s recommendations [37].

Owing to a rapidly aging population, a phenomenon that is common to many countries worldwide, South Korea is expected to have more than 30% of its population aged 65 years or older by 2036. Furthermore, the speed with which the Korean population is aging is faster than that in other developed countries [8,48]. Facing the increasing burden of an elderly population will be particularly challenging in emerging markets where such demographic transition is far more rapid compared to most developed countries [49]. Such a scenario predicts that the future elderly patients’ OOP expenditure will be high because of an increased number of outpatient visits. Thus, there is a need for continuous evaluation of the impact of the fixed copayment model on healthcare utilization and OOP expenditure among the Korean elderly.

Our results indicated that the influence of cost sharing differs according to patients’ income levels. Moreover, reduced copayments increased healthcare utilization exclusively among the Korean elderly with low income. Therefore, lower copayment models or changes in coinsurance rates can be effective tools for enhancing the accessibility of healthcare services and prescription drugs to patients with lower incomes.

Our study has several strengths. First, we have identified how healthcare utilization among elderly Korean patients is influenced by policy changes in the cost sharing of the Korean NHI system using individual longitudinal data. Second, we examined the effects of OOP expenditure on healthcare utilization according to income status.

Nevertheless, several limitations need to be considered when interpreting our results. First, we analyzed healthcare utilization data before and after the age of 65, which refers to the period when the copayment model begins to be implemented in elderly Korean patients. However, reduced copayments can influence healthcare utilization in different ways among those who are very elderly [31]. Second, because the copayment model is applied to all elderly Korean patients covered by the NHI program, approximately 97% of Korean citizens [16], we were unable to obtain a control group with a comparable age to that of the case group. For that reason, patients aged 64 years or younger were selected as the control group. It is also important to highlight that there were no significant between-group differences in educational attainment or the number of chronic diseases. Although the case group was older than the control group by one year, there was no significant difference in the number of chronic diseases between the two groups. However, the possibility of potentially unobserved biological differences cannot be excluded. Third, not only patients but also clinicians influence healthcare utilization. We were not able to include the doctors’ characteristics as control variables because of the lack of available data on the subject. Fourth, a one year analysis does not provide sufficient time to identify concrete behavioral changes. Therefore, further research is needed, as increased outpatient visits due to reduced cost sharing may produce mixed outcomes. While it may improve healthcare utilization, it may also increase the moral hazard behavior of patients and physicians. Finally, although we were unable to analyze the effects of changes in healthcare utilization—owing to the cost sharing policy changes—on patients’ health (because it was beyond the scope of the current paper), this remains an important topic for future research endeavors.

## 5. Conclusions

Many Asian countries are confronted with rapidly aging populations, including South Korea. As of 2018, approximately 14% of the total Korean population comprised older adults aged 65 years and above [48]. This study showed that there was an increase in healthcare service utilization among the poorer Korean elderly because of cost reduction via implementation of the lower copayment model. Our results provide insight into payment policies for healthcare insurances in aging or aged societies in which the elders’ income level may be lower than that of the general population owing to reduced pension benefits. Governments in middle or high income countries with universal insurance coverage can control healthcare costs and develop a sustainable financial system by modifying cost sharing for the elderly.

## Figures and Tables

**Table 1 ijerph-17-08118-t001:** Participants’ sociodemographic characteristics by group.

Variable	Case Group (*n* = 558)	Control Group (*n* = 587)	*p*-Value ^1^
Sex, *n* (%)			
male	251 (45.0)	256 (43.6)	0.6407
female	307 (55.0)	331 (56.4)	
Residential area, *n* (%)			
metropolitan	246 (44.1)	243 (41.4)	0.3578
non-metropolitan	312 (55.9)	344 (58.6)	
Educational attainment, *n* (%)			
elementary school or lower	243 (43.6)	236 (40.2)	0.5182
middle school	121 (21.7)	135 (23.0)	
high school or higher	194 (34.8)	216 (36.8)	
Household income, *n* (%) ^2^			
tercile 1 (low)	194 (34.8)	181 (30.8)	0.0436
tercile 2 (middle)	193 (34.6)	185 (31.5)	
tercile 3 (high)	171 (30.7)	221 (37.7)	
Number of chronic diseases, *n* (%)			
0–1	148 (26.5)	174 (29.6)	0.2231
2	102 (18.3)	119 (20.3)	
≥3	308 (55.2)	294 (50.1)	
Smoking status, *n* (%)			
current smoker	89 (16.0)	90 (15.3)	0.9043
ex-smoker	120 (21.5)	132 (22.5)	
non-smoker	349 (62.5)	365 (62.2)	
Survey year, *n* (%)			
2011	194 (34.8)	203 (34.6)	0.7807
2013	167 (29.9)	186 (31.7)	
2015	197 (35.3)	198 (33.7)	

^1^ Chi-square test. ^2^ Household income was classified into three groups each year according to the household equivalent income: tercile 1 (low: 141–1253, 250–1340, and 297–1445 thousand won in 2011, 2013, and 2015); tercile 2 (middle: 1256–2100, 1341–2321, and 1446–2478 thousand won in 2011, 2013, and 2015); and tercile 3 (high: 2112–8814, 2333–13,808, and 2489–16,329 thousand won in 2011, 2013, and 2015).

**Table 2 ijerph-17-08118-t002:** Outpatient clinic visit frequency, total prescriptions, and out-of-pocket (OOP) expenditure during one year before and after the cost sharing change by group.

Variable	Case Group			Control Group		
	Pre-65	Post-65	Pre-Post Difference	Pre-64	Post-64	Pre-Post Difference
**Number of patients who visited a clinic at least once, *n* (%)**						
total	501 (89.8)	499 (89.4)	−2 (−0.4)	506 (86.2)	527 (89.8)	21 (3.6) **
low income	173 (89.2)	171 (88.1)	−2 (−1.1)	163 (90.1)	167 (92.3)	4 (2.2)
middle income	169 (87.6)	172 (89.1)	3 (1.5)	163 (88.1)	161 (87.0)	−2 (−1.1)
high income	159 (93.0)	156 (91.2)	−3 (−1.8)	180 (81.5)	199 (90.1)	19 (8.5) ***
**Number of o** **utpatient visits** **, mean (SE)**						
total	18.6 (0.8)	20.9 (0.9)	2.3 (0.6) **	17.3 (0.9)	17.8 (0.9)	0.4 (0.7)
low income	18.5 (1.3)	22.3 (1.5)	3.9 (1.1) ***	19.1 (1.6)	18.4 (1.4)	−0.7 (1.0)
middle income	17.8 (1.1)	19.0 (1.3)	1.2 (1.1)	17.4 (1.8)	16.6 (1.6)	−0.8 (1.1)
high income	19.7 (1.7)	21.4 (2.0)	1.7 (1.4)	15.9 (1.3)	18.3 (1.7)	2.4 (1.4) *
**Number of prescription drugs, mean (SE)**						
total	11.6 (0.5)	12.5 (0.5)	-0.9 (0.3) **	10.3 (0.5)	10.5 (0.4)	0.2 (0.3)
low income	11.9 (0.7)	13.3 (0.9)	1.4 (0.6) **	11.4 (0.8)	11.8 (0.7)	0.4 (0.6)
middle income	11.4 (0.7)	12.2 (0.8)	0.8 (0.6)	10.0 (0.8)	9.8 (0.7)	−0.2 (0.5)
high income	11.5 (0.7)	11.9 (0.9)	0.4 (0.5)	9.6(0.7)	10.0 (0.7)	0.4 (0.5)
**OOP o** **utpatient visit expenditure** **, mean (SE) ^1^**						
total	427.1 (41.9)	378.7 (36.7)	-42.3 (52.2)	392.0 (37.5)	381.6 (35.0)	−6.9 (50.2)
low income	453.7 (92.6)	369.4 (59.6)	−73.2 (107.2)	329.7 (44.9)	305.2 (55.3)	−25.6 (66.1)
middle income	416.9 (61.3)	365.3 (64.2)	−40.6 (82.0)	478.2 (85.8)	342.0 (52.1)	−137.4 (104.7)
high income	409.1 (55.7)	403.5 (66.9)	−11.2 (78.2)	370.4 (58.2)	477.8 (67.9)	126.2 (85.6)
**OOP prescription drug** **expenditure** **, mean (SE) ^1^**						
total	119.0 (6.3)	120.0 (6.2)	3.1 (4.4)	113.0 (5.6)	116.2 (5.6)	4.4 (3.9)
low income	124.1 (12.7)	135.5 (12.2)	108.6 (10.3)	111.4 (8.9)	121.1 (9.5)	11.9 (8.0)
middle income	110.6 (9.0)	107.0 (8.7)	−0.5 (72.5)	105.0 (9.2)	112.2 (9.8)	2.4 (4.8)
high income	122.3 (10.6)	118.1 (10.8)	−1.8 (5.6)	121.7 (10.7)	115.4 (9.8)	−0.2 (7.0)

^1^ 1000 won.* *p* < 0.1; ** *p* < 0.05; *** *p* < 0.01. Abbreviations: OOP, out-of-pocket; SE, standard error.

**Table 3 ijerph-17-08118-t003:** Results of the difference-in-differences (DID) analysis ^1^ for outpatient clinic visit frequency, total prescriptions, OOP outpatient visit expenditure, and OOP prescription drug expenditure by group and time.

Variable	Outpatient Clinic Visit Frequency, Beta (SE)	Total Prescriptions, Beta (SE)	OOP Outpatient Visit Expenditure, Beta (SE)	OOP Prescription Drug Expenditure, Beta (SE)
Group				
case	0.70 (1.19)	1.00 (0.61) *	36.30 (53.59)	3.33 (7.84)
control	Ref.	Ref.	Ref.	Ref.
Time				
post-	0.44 (0.68)	0.21 (0.32)	−8.01 (48.91)	4.57 (4.13)
pre-	Ref.	Ref.	Ref.	Ref.
Case group × post-time	1.83 (0.98) *	0.67 (0.45)	−39.03 (69.64)	−1.89 (5.83)

^1^ Adjustments were made for sex, residential area, educational attainment, household equivalent income, the number of chronic diseases, smoking status, and survey year.* *p* < 0.1. Abbreviations: DID, difference-in-differences; OOP, out-of-pocket; Ref., reference; SE, standard error.

**Table 4 ijerph-17-08118-t004:** Results of the DID analysis ^1^ for outpatient clinic visit frequency, total prescriptions, OOP outpatient visit expenditure, and OOP prescription drug expenditure by household income.

Variable	Outpatient Clinic Visit Frequency, Beta (SE)	Total Prescriptions, Beta (SE)	OOP Outpatient Visit Expenditure, Beta (SE)	OOP Prescription Drug Expenditure, Beta (SE)
Low income (*n* = 375)				
case group (ref.: control group)	−1.10 (1.93)	0.45 (1.08)	125.32 (94.91)	12.76 (15.08)
post-time (ref.: pre-time)	−0.72 (1.09)	0.44 (0.58)	−21.25 (88.03)	11.87 (9.49)
case group × post-time	4.58 (1.51) ***	0.98 (0.80)	−59.74 (123.17)	−1.52 (13.06)
Middle income (*n* = 378)				
case group (ref.: control group)	0.37 (1.99)	1.18 (1.05)	−27.17 (94.37)	−1.22 (12.26)
post-time (ref.: pre-time)	−0.77 (1.08)	−0.24 (0.56)	−132.75 (90.66)	3.46 (5.43)
case group × post-time	1.94 (1.51)	1.04 (0.78)	83.58 (126.60)	−4.48 (7.54)
High income (*n* = 392)				
case group (ref.: control group)	3.34 (2.29)	1.63 (1.05)	37.93 (90.45)	−0.10 (13.77)
post-time (ref.: pre-time)	2.40 (1.32) *	0.41 (0.52)	108.30 (76.33)	−0.73 (6.11)
case group × post-time	−0.71 (2.00)	−0.03 (0.78)	−117.42 (112.69)	−0.95 (9.02)

^1^ Adjustments were made for sex, residential area, educational attainment, household equivalent income, the number of chronic diseases, smoking status, and survey year.* *p* < 0.1; *** *p* < 0.01. Abbreviations: DID, difference-in-differences; OOP, out-of-pocket; Ref., reference; SE, standard error.

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
