# Peer review of "Who Benefits from the Fixed Copayment of Medical and Pharmaceutical Expenditure among the Korean Elderly?"

_ijerph, 2020, doi:10.3390/ijerph17218118_

Round 1

Reviewer 1 Report

This is very valuable contribution on South Korean health system perspective on Fixed Copayments of Medical and Pharmaceutical Expenditure Among the Elderly Citizens.

It is also interesting in the context to obtain transnational comparison with the surrounding Asian systems.

Its only minor bottleneck weakness is to diverisy evidence base which is so far too much leaning towards the Korean national and OECD academic sources.

Far more evidence outsourcing from the LMICs nations and Emergign Markets should be cited to support claims in the Introductory and Discussion sections.

Such increased heterogeneity of citation track record would give additional credibility to the study.

My proposed sources to be included alongside few ones at authors own disposal are the following ones:

https://www.frontiersin.org/articles/10.3389/fphar.2018.00960/full

https://academic.oup.com/ije/article/46/2/e15/2617147

https://resource-allocation.biomedcentral.com/articles/10.1186/s12962-020-00210-2

https://academic.oup.com/heapol/article/24/1/63/598886

https://synapse.koreamed.org/DOIx.php?id=10.4093/jkda.2007.31.4.362&vmode=FULL

https://bmjopen.bmj.com/content/7/9/e016640.abstract

https://apps.who.int/iris/handle/10665/330337

https://academic.oup.com/ije/article/46/3/799/2418193

https://www.tandfonline.com/doi/abs/10.1080/13696998.2019.1600523

https://www.tandfonline.com/doi/abs/10.3111/13696998.2015.1093493

https://www.frontiersin.org/articles/10.3389/fpubh.2015.00135/full

https://www.ncbi.nlm.nih.gov/pmc/articles/PMC6835015/

https://www.frontiersin.org/articles/10.3389/fphar.2016.00021/full

Conditional to authors adopting at least several of the aforementioned sources I would be willing to review the revised manuscript assuming its maturity for publishing.

Reviewer 2 Report

To Dear Authors,

My suggestions are as follows,

This manuscript by Eunja and co-workers deals with the Fixed Copayment of Medical and Pharmaceutical Expenditure Among the Elderly in Korea. It is an interesting issue that who benefits from the Korea National Health Insurance; however, there are some unclear parts in this manuscript. First, the age was grouped as less than 65 and more than 65, can you provide more accurate demographic distribution of your samples? Second, can you tell us the amount of income is low, middle, high in Korea?

I believe this manuscript need more improvement to be accepted, and it will be contributed to addictive behaviors and this field a lot in the future.

Best regards,

Reviewer 3 Report

This study examines effects of a lower co-payment care model on various health consumption measure in Korea. The paper shows a comprehensive investigation of this critical issue. I have only a few questions and comments as follows

  1. I am surprise to see that the authors did not apply any panel data methods (e.g., fixed effects or random effects) to control for individual unobserved characteristics despite they have access to a panel data set. Please explain why?
  2. In section 2.1 the authors stated that the KHP was conducted annually since 2008 but I’m not sure why they analyse only three odd years (2011, 2013, 2015) while obtaining data from 2010-2016. Why data in even year (2010, 2012, 2014, 2016 were not used?
  3. The definition of “pre”, “post”, “case” and “control” are not clear. Ideally “case” and “control” should have similar characteristics but in this case those eligible for the program (age 65=+) and those ineligible (age <65) are biologically differed. I believe that people aged 65+ will consume health services more than those in the younger group. So these two groups are not ideal “case” and “control”. Also, the authors have data from 2010-2016 while the program implemented since 2007 (line 65), so there’s no “pre” and “post” program. I suggest the author define clearly “pre” and “post” refer to pre-age post age 65. Mean while the “control” group, I believe are those who never turn 65 in the study period (e.g., those age <65 in 2016). Please explain the group clearly if my guess are not correct. Also, make caveat about this second-best choice of difference in difference. This observation may require the authors to put emphasis on potential treatments for unobserved heterogeneity (e.g. fixed vs random effects, instrumental variables, or robustness check such as “e-value”).
  4. The authors report a lot of covariates were controlled for but no covariates were reported in regression results. I’m curious why?
  5. Conducting separate regressions for each income group incur a cost of losing observations. The authors can create dummies variable to represent income groups and use all observations of the whole panel in analysis. The added benefit of this choice is that they don’t need to face the difficult question that parameters generated from samples of different populations (e.g. income group) are not directly comparable.
  6. The introduction is quite lengthy. They may need just three paragraph: 1) the importance of problem? 2) the gap in the literature, and 3) contribution of this paper.

Round 2

Reviewer 3 Report

The response addressed my questions well.